# MRI and CT Fusion in Stereotactic Electroencephalography (SEEG)

**DOI:** 10.3390/diagnostics13223420

**Published:** 2023-11-09

**Authors:** Jaime Pérez Hinestroza, Claudia Mazo, Maria Trujillo, Alejandro Herrera

**Affiliations:** 1Multimedia and Computer Vision Group, Universidad del Valle, Cali 760042, Colombia; claudia.mazovargas@ucd.ie (C.M.); maria.trujillo@correounivalle.edu.co (M.T.); herrera.alejandro@correounivalle.edu.co (A.H.); 2School of Computing, Faculty of Engineering and Computing, Glasnevin Campus, Dublin City University, 9 Dublin, Ireland; 3Clinica Imbanaco Grupo Quironsalud, Cali 760042, Colombia

**Keywords:** image fusion, stereotactic electroencephalography, computer tomography, magnetic resonance imaging, image registration

## Abstract

Epilepsy is a neurological disorder characterized by spontaneous recurrent seizures. While 20% to 30% of epilepsy cases are untreatable with Anti-Epileptic Drugs, some of these cases can be addressed through surgical intervention. The success of such interventions greatly depends on accurately locating the epileptogenic tissue, a task achieved using diagnostic techniques like Stereotactic Electroencephalography (SEEG). SEEG utilizes multi-modal fusion to aid in electrode localization, using pre-surgical resonance and post-surgical computer tomography images as inputs. To ensure the absence of artifacts or misregistrations in the resultant images, a fusion method that accounts for electrode presence is required. We proposed an image fusion method in SEEG that incorporates electrode segmentation from computed tomography as a sampling mask during registration to address the fusion problem in SEEG. The method was validated using eight image pairs from the Retrospective Image Registration Evaluation Project (RIRE). After establishing a reference registration for the MRI and identifying eight points, we assessed the method’s efficacy by comparing the Euclidean distances between these reference points and those derived using registration with a sampling mask. The results showed that the proposed method yielded a similar average error to the registration without a sampling mask, but reduced the dispersion of the error, with a standard deviation of 0.86 when a mask was used and 5.25 when no mask was used.

## 1. Introduction

Epilepsy is a neurological disorder with a worldwide prevalence of 0.8% to 1.2%, where 20% to 30% of cases are untreatable with Anti-Epileptic Drugs (AED) [1,2]. For those patients, a valuable treatment is a surgical intervention [3], with a success rate ranging from 30% to 70% [4].

The success of the surgical intervention depends on a precise localization of the epileptogenic tissue. Diagnostic techniques, including but not limited to Stereotactic Electroencephalography (SEEG), play a crucial role in achieving this accuracy [3,5,6]. The SEEG measures the electric signal within the brain areas using deep electrodes, guiding the implantation and electrode localization with Magnetic Resonance Imaging (MRI) and Computer Tomography (CT) images. However, given the limited structural details in CT images, a fusion with an MRI is required. This fusion ensures a comprehensive representation of both anatomical structures and electrode positions in a unified image [7,8,9].

Image fusion is a processing technique that involves mapping images into a common coordinate system and merging the aligned results into a single output. Numerous methods are available for image fusion; however, the performance of each technique is influenced by characteristics related to the acquisition and image type [10,11]. When external objects are present in an SEEG sequence, it may interfere with the registration process, which relies on similarity metrics computed using voxel data between images [10,11]. Consequently, changes in the structural data of the CT image can affect the calculation of similarity metrics, leading to misregistration in the fused images.

Based on the challenges associated with image registration, we conducted a systematic review, using the methodology outlined by Kitchenham [12] for literature reviews in software engineering. Our research looked into the techniques and tools used for brain image fusion between CT and MRI, as well as the validation techniques employed to measure the performance [13]. Our review revealed a notable absence of a standard method for image fusion validation in CT and MRI, especially when external objects are present. Furthermore, we identified a significant lack of validation methodologies for these techniques. This is particularly concerning given that the Retrospective Image Registration Evaluation Project (R.I.R.E.), once a standard methodology, is no longer in use. Our review also highlighted the importance of understanding the performance of various image fusion techniques in applications like SEEG that involve external objects. We found that methods using Mutual Information (MI) as the optimization metric exhibited superior performance in multimodal image fusion.

These challenges were evident in SEEG examinations conducted at Clinica Imbanaco Grupo Quironsalud in Cali, Colombia, where we identified registration errors in the fusion of MRI and CT images primarily attributed to the presence of electrodes. These inconsistencies required manual adjustments to correctly align the misregistered MRI images. In response to these challenges, we introduce an image fusion method that accounts for external elements, primarily in exams like SEEG. It is crucial to note that while our method is designed to mitigate the impact of external objects in the images and enhance the spatial accuracy of electrode localization, it does not explore into the analysis of electric signals from the deep electrodes. Such an analysis falls outside the scope of this study and would demand a distinct analytical framework.

## 2. Fusion Method

The general procedure shown in Figure 1 consists of seven main steps: (i) initial electrode segmentation in the CT image; (ii) generation of a mask of all non-electrode voxels in the CT image; (iii) registration of the MRI against the CT image using the non-electrode sampling mask to compute the transformation; (iv) segmentation of the brain from the registered MRI with the ROBEX tool, and subsequently computing a brain mask; (v) improving the electrode segmentation using the brain mask obtained from the previous step; (vi) integrate the fully segmented electrodes with the registered MRI.

### 2.1. CT Electrode Segmentation

Given that our literature search did not identify any fusion methods that take into account the electrode and utilize segmentation for registration [13], we devised our own segmentation procedure. This procedure employs thresholding and morphological operations, as shown in Figure 2 and Figure 3.

To extract the electrodes from the CT, we employed simple thresholding with a window of 1500 HU to 3000 HU (i). Subsequently, we computed a mask of the head tissue to remove the skull from the segmented image (viii). Next, we generated a brain mask for the MRI, which had been aligned with the CT, utilizing the ROBEX (Robust Brain Extraction) tool, which is a stripping method based on the work of Iglesias et al. [14]. Finally, we applied this mask to the registered electrodes to remove any objects situated outside the brain.

To segment the skull, we employed a simple threshold with a window ranging from 300 HU to 1900 HU (ii). Subsequently, we employed a morphological eroding technique with a cross kernel of size 3×3×3 to eliminate the electrodes from the skull. Next, a morphological closing and dilation operation with a ball kernel of size 4×4×4 was performed to connect all the bone tissue (iii). Finally, we applied a NOT operator to generate a mask of no-skull gray voxels (iv).

We generate a head mask using Otsu’s thresholding (v) to exclude any object external to the head [15]. After that, we apply a morphological hole-filling operation to remove any internal gap within the head (vi). Then, we create a brain mask by intersecting the head mask and the no-skull mask (vii). Finally, we combine the brain mask with the thresholding electrodes, ensuring the removal of most of the skull tissue (viii).

### 2.2. MRI Registration

For MRI registration, we employed an affine rigid transformation combined with a gradient descent algorithm, using Mutual Information (MI) as the similarity metric. We opted for this registration approach because MI is based on the normal probability distribution between images [10,11], which has been shown to be more effective in multi-modal registration [13]. To enhance the registration process, we add a unique step that uses a sampling of the voxels that do not contain electrodes when computing the MI. We achieve this by creating a mask through the application of a NOT operation to the segmented electrodes, as detailed in Section 2.1. Figure 4 shows a schema of the registration procedure.

### 2.3. Final CT Electrode Segmentation

The preliminary electrode segmentation employs both thresholding and morphological operations. However, this approach might segment some bone tissue alongside the electrodes, which must be excluded from the final image. To address this, we use the aligned MRI to generate a brain mask with the aid of the ROBEX tool, as depicted Figure 5.

### 2.4. Image Merging

Finally, we add the segmented electrodes to the aligned MRI to produce the fused image (Figure 6).

## 3. Validation Method

Given the lack of a standardized validation methodology for multi-modal image fusion, as highlighted in our 2021 literature review [13], we opted to employ two distinct methods to validate our proposed technique. Initially, we employ the RIRE dataset to generate synthetic data. Then, for our second validation, we used four pairs of MRI and CT images from SEEG exams, measuring the performance by identifying five anatomical structures in the CT and MRI.

### 3.1. Validation Using RIRE Dataset

We selected eight images from the RIRE dataset containing both MRI and CT images. To simulate the presence of electrodes, we introduced cylinders into the CT images. Subsequently, we performed a rigid registration on the images without electrodes. The transformation obtained from this registration served as a reference for further analysis.

Furthermore, to measure the performance, we compared the location of brain structures in the registered images. This comparison was conducted using the Euclidean distance between the reference structure in the CT and the corresponding structure in the registered MRI.

#### 3.1.1. Electrodes Generation

To obtain a CT image with electrodes, we used the RIRE dataset and added cylinders to the images to simulate SEEG electrodes. These simulated electrodes were designed with a diameter of 3 mm and a length of 80 mm, mirroring the specifications of deep electrodes that feature eight contacts with a 10 mm spacing between them [16]. The gray values of the generated electrodes ranged between 1500 and 3000 HU. In total, we added a total of 12 electrodes to the CT images, placing them at random orientations and positions within the brain tissue. Figure 7 shows an example of the generated images.

#### 3.1.2. Reference Registration Images

To evaluate the performance of our method, we applied a rigid registration on the RIRE dataset without electrodes. This registration procedure employed an affine transformation with a gradient descent algorithm. We used the eight points defined in the RIRE dataset to calculate the reference point using the reference registration. Table 1 presents the original points used, while Table 2 displays the resulting reference points per image.

#### 3.1.3. Registration Error Using Reference Points

We computed the error using the resulting points in Table 2 and compared them to the resulting point from the procedure described in Section 2. The error was calculated as the Euclidean distance between the reference points and the points obtained from our fusion method.
(1)$$error=\sqrt{{\left(PR_{x}-P_{x}\right)}^{2}+{\left(PR_{y}-P_{y}\right)}^{2}+{\left(PR_{z}-P_{z}\right)}^{2}},$$
where $PR_{x},PR_{y},PR_{z}$ are the coordinates in mm for the reference points (Table 2), and $P_{x},P_{y},P_{z}$ are the reference points per image after applying our fusion method.

### 3.2. Validation with Brain Structures

We measure the performance of the fusion method by using the subsequent brain structures in the CT image as reference points: (i) the Sylvian aqueduct; (ii) the anterior commissure; (iii) the pineal gland; (iv) the right lens; (v) the left lens.

With the guidance of a medical expert, we manually localized the structures of interest, obtaining their positions in both the CT and registered MRI images using the 3D Slicer version 4.11. Upon identifying these structures, we measured the error as the Euclidean distance between the reference structure point in the CT image and the corresponding point in the registered MRI using Figure 1. For performance evaluation, we evaluated the error in images resulting from two distinct methods: (i) our proposed approach that incorporates a sampling mask during registration, and (ii) a reference method from the existing literature that conducts registration without a sampling mask. The purpose of this validation was to determine whether the use of a mask reduces registration errors. Figure 8 and Figure 9 show the methods that were compared.

In our validation process, we also employed global fusion metrics to assess potential distortions arising from the fusion procedures. The metrics we utilized include:
Mutual Information (*MI*):

Estimate the amount of information transferred from the source image into the fused image [17]. Given the input images ($I_{i}$) and the fused image $I_{f}$, the MI can be computed using the following equation:(2)$$MI\left(I_{i},I_{f}\right)=H\left(I_{i}\right)+H\left(I_{f}\right)-H\left(I_{i},I_{f}\right),$$
where $H(I_{i},I_{f})$ is the joint entropy between the input and fused images, and $H(I_{i})$, $H(i_{f})$ are the marginal entropy of the input and fused image, respectively.

Structural Similarity Index (*SSIM*):

Measure the preservation of the structural information, separating the image into three components: luminance *I*, contrast *C* and structure *S* [17].
(3)$$SSIM\left(I_{i},I_{f}\right)={\left[I\left(I_{i},I_{f}\right)\right]}^{{}^{a}}.{\left[C\left(I_{i},I_{f}\right)\right]}^{{}^{b}}.{\left[S\left(I_{i},I_{f}\right)\right]}^{{}^{c}}.$$

Root Mean Square Error (*RMSE*):

Measure the variance of the arithmetic square root [17].
(4)$$RMSE=\sqrt{\left(\sum_{x=1}^{M}\sum_{y=1}^{N}{\left[I_{i}\left(x,y\right)-I_{f}\left(X,Y\right)\right]}^{2}\right)},$$
where $I_{i}\left(x,y\right)$ and $I_{f}\left(X,Y\right)$, are the pixel values of the input and fused image, respectively, and *M*, *N* are the dimensions of the image.

Peak Signal-to-Noise Ratio (*PSNR*):

The PSNR is calculated from the RMSE in the following equation given image of dimension M×N [17]
(5)$$PSNR=10\;\cdot \;Log\left[\frac{{\left(M\times N\right)}^{{}^{2}}}{RMSE}\right].$$

## 4. Results

We validated the performance of the fusion method following the two methodologies in Section 3. We used eight pairs of CT and MRI from the RIRE dataset for the first method. The CT images were generated with the method described in Section 3.1.1. We compared the procedure shown in Figure 8 against a fusion procedure that does not employ a sampling mask of the brain tissue, which is shown in Figure 9. Both methods used a rigid registration with MI as the similarity metric and gradient descent for the optimization. We used descriptive statistical metrics of central tendency and variation to compare the methods using the validation from Section 3.1. These results were summarized in the box plot shown in Figure 10. For the second validation, we faced a limitation in the number of images available for evaluation. Given this constraint, we opted to compare the methods individually for each of the four cases. A scatter plot was chosen as the most suitable representation to visualize the error dispersion for both methods. Scatter plots are particularly effective in such scenarios as they allow for clear visualization of individual data points, making it easier to discern patterns or anomalies, especially when dealing with a smaller dataset. This approach provides a more transparent and detailed view of the distribution of errors across the limited set of images. The results of this comparison are illustrated in Figure 11.

Table 3 displays the Euclidean distance between the reference points and the resulting points of the transformation from the compared methods. From the data, we can observe that the difference in the Euclidean distance for our method is significantly lower in images 3, 6, and 8. This is mainly caused by the differences in the original images that have some variations in brain tissue, as shown in Figure 12, Figure 13 and Figure 14. Due to some electrodes passing through these areas with variations, the sampling in the registration does not use these voxels to compute the transformation, thus improving the registration when the mask is used. The results are represented in Figure 10, where our method using a sampling mask yields a Euclidean distance of 1.3176 mm with a standard deviation of 0.8643. In contrast, the method without a sampling mask yields a Euclidean distance of 1.2789 mm with a standard deviation of 5.2511. These findings suggest that the use of the mask improves the registration when there is a great difference in the tissue between the MRI and CT images due to the reduction in voxel sampling of these varying tissues in the registration process.

For the second validation, we use the methodology described in Section 3.2 in four pairs of MRI and CT images obtained from Clínica Imbanaco Grupo Quirón Salud. Table 4 displays the localized points in the CT image that we used as a reference in our analysis. The localized points for the structure in the MRI registered with our proposed methods are shown in Table 5, while the points in the MRI from the method to compare are displayed in Table 6.

After the structure localization, we compute the Euclidean distance between the points from the registered MRI against the points of the CT images. This process is applied the resulting images from our proposed method and the method that uses no sampling mask in the registration. The resulting Euclidean distances are shown in Table 7.

The validation results, displayed in Table 7, show that our method has a higher Euclidean distance compared to the method without a mask for images from patient 1, patient 3, and patient 4. However, our method achieves a lower Euclidean distance in images from patient 2. Further analysis of the two methods using global performance metrics, as presented in Table 8, reveals a relatively low difference, indicating minimal distortion in the images when comparing the two methods.

While the validation with the limited dataset showed some promising results, it still requires further refinement. While we were able to reduce the Euclidean distances for all structures in patient 2’s images and for some structures, such as the anterior commissure and the pineal gland, in patient 3’s images, our method displayed a higher Euclidean distance in images for patient 1 and patient 4. We employed global fusion metrics from Table 8 to analyze if the difference in distance was caused by any distortion in the registration procedure. However, these metrics did not reveal any significant difference related to distortion in the registered images using any of the compared registration methods.

While the application of the mask did induce an increase in error for certain images, the implementation of our method with the mask notably reduced the average error and overall dispersion, as depicted in Figure 11. This demonstrates the promising potential of our method. However, with the limited dataset, while showing some promising results, it is evident that further refinement is necessary. To conclusively affirm the improvement introduced by our approach, a larger dataset is required for this validation methodology.

## 5. Discussion

Our proposed image fusion method between MRI and CT, which considers the electrodes, is useful in addressing the identified problem, where the presence of external objects produced a registration error. This approach improves the registration in the images using the RIRE dataset. However, in the second validation stage, our method demonstrated a lower average error, yet we observed instances where performance was lower when the sampling mask was applied. This could be attributed to the potential importance of information proximal to the electrode for the calculation of similarity metrics during the registration procedure. However, even in these cases, the error increase was minimal compared to scenarios where the error was lower.

Another challenge was the absence of a standardized validation method for multimodal fusion, prompting us to develop our own method using available data. This included the limited dataset from Imbanaco and the deprecated RIRE dataset.

We also found a lack of research on methods for electrode segmentation in SEEG. This necessitated the development of our own registration method, a task that was not initially included in the project scope.

All objectives of the project were achieved. The project successfully identified primary techniques for image registration and fusion between MRI and CT images, developed a method to fuse these images when external objects were present, and conducted an evaluation to measure and compare the performance of the designed method. In the evaluations, the method outperformed other existing state-of-the-art techniques in certain scenarios.

## 6. Conclusions

We have developed and presented an image fusion method for combining CT and MRI from SEEG exams. Our approach aims to minimize misregistration errors between pre-surgical MRI and post-surgical CT images, by the use of a sampling mask of all voxels that are not electrode in the post-surgical CT image.

We acknowledged the lack of a standard validation method for image fusion and registration in brain images, mainly when external objects are presented in one of the images. We addressed this by employing two evaluation approaches: (i) a simulation-based evaluation method with synthetic electrodes generated from the RIRE dataset; and (ii) an evaluation using four image pairs acquired from patients at Clinica Imbanaco Grupo Quirón Salud, where we measured the error using five anatomical structures that can be localized in the pre-surgical MRI and post-surgical CT images.

Our findings indicate that the proposed method outperforms the existing state-of-the-art techniques in the simulation-based evaluation using the RIRE dataset. In the evaluation using clinical images, we observed that our method demonstrated superior performance in some cases, while showing a slight decrease in performance in others. Despite this variability, the overall average Euclidean distance was lower for our method, suggesting an improvement in registration accuracy.

We recommend enhancing the second validation methodology by increasing the number of images and refining the localization of brain structures to further reduce bias in the evaluation results. This would enable a more comprehensive assessment of the proposed fusion method’s performance for clinical scenarios.

In conclusion, our proposed image fusion method shows promise for improving the accuracy of the registration in SEEG. With further development and refinement, this approach has the potential to significantly impact the field of epilepsy treatment, offering further aid in the localization of epileptogenic tissue when SEEG is employed.

## Figures and Tables

**Figure 1 diagnostics-13-03420-f001:**
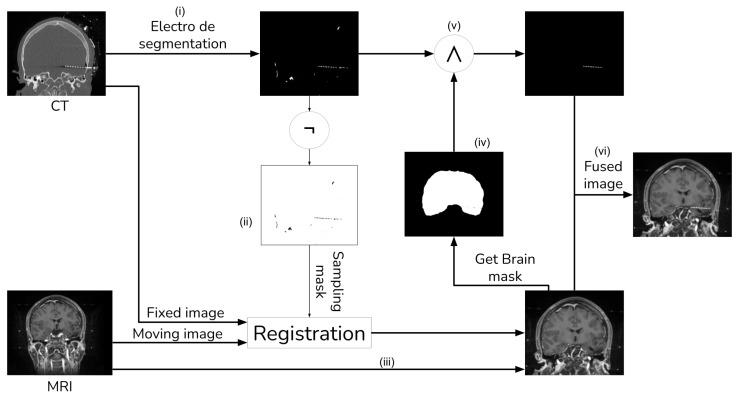
Image fusion method with external object.

**Figure 2 diagnostics-13-03420-f002:**
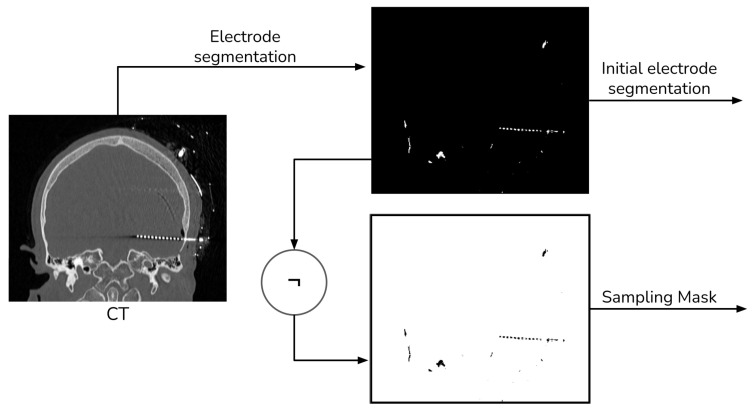
Electrode segmentation general procedure.

**Figure 3 diagnostics-13-03420-f003:**
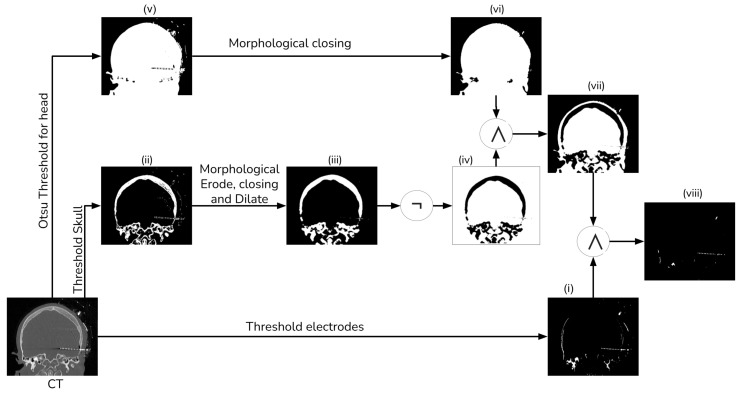
Electrode segmentation detailed procedure.

**Figure 4 diagnostics-13-03420-f004:**
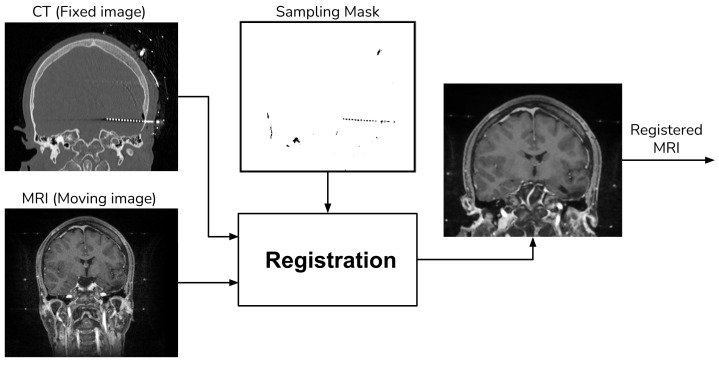
Registration procedure.

**Figure 5 diagnostics-13-03420-f005:**
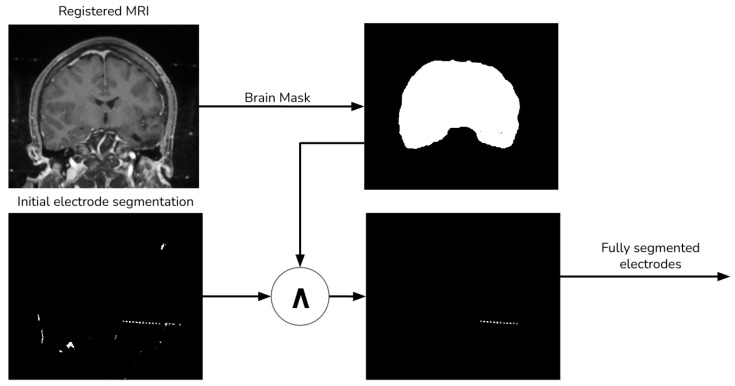
Final electrode procedure.

**Figure 6 diagnostics-13-03420-f006:**
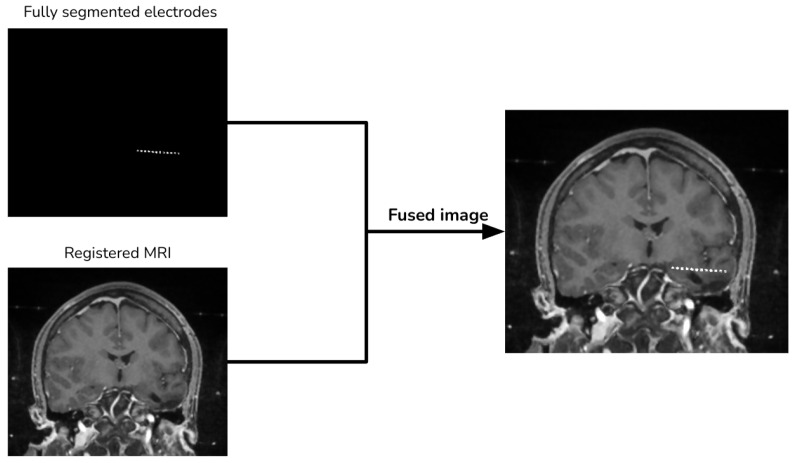
Image merging procedure.

**Figure 7 diagnostics-13-03420-f007:**
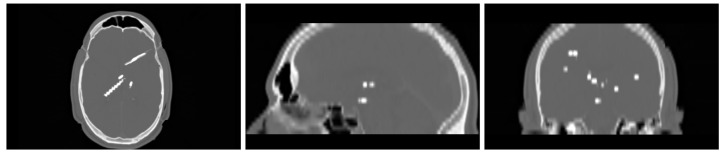
Example of a generated image with electrodes.

**Figure 8 diagnostics-13-03420-f008:**
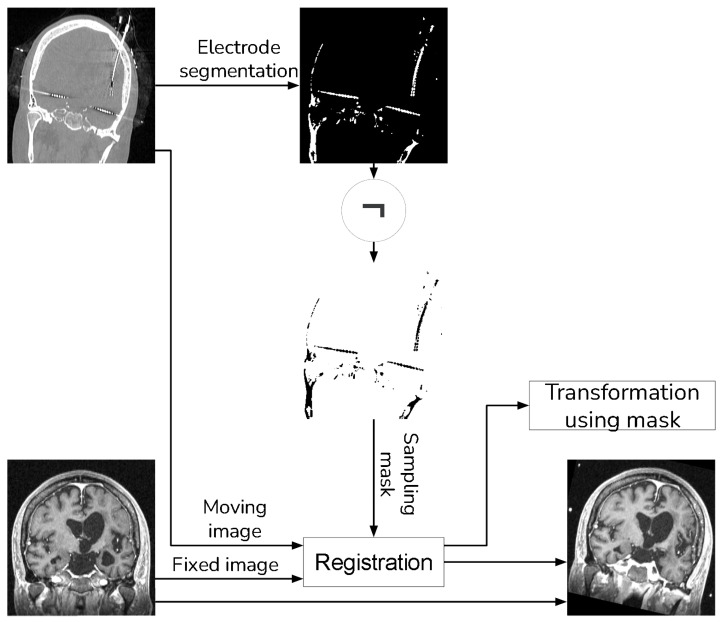
Proposed method that uses a sampling mask for the registration.

**Figure 9 diagnostics-13-03420-f009:**
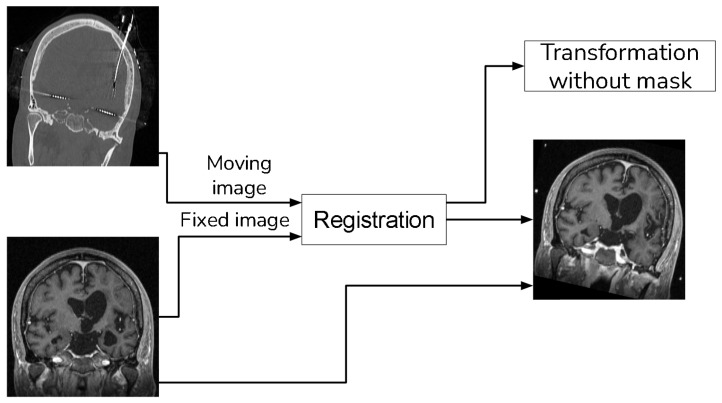
Method to compare that does not use a sampling mask in the registration.

**Figure 10 diagnostics-13-03420-f010:**
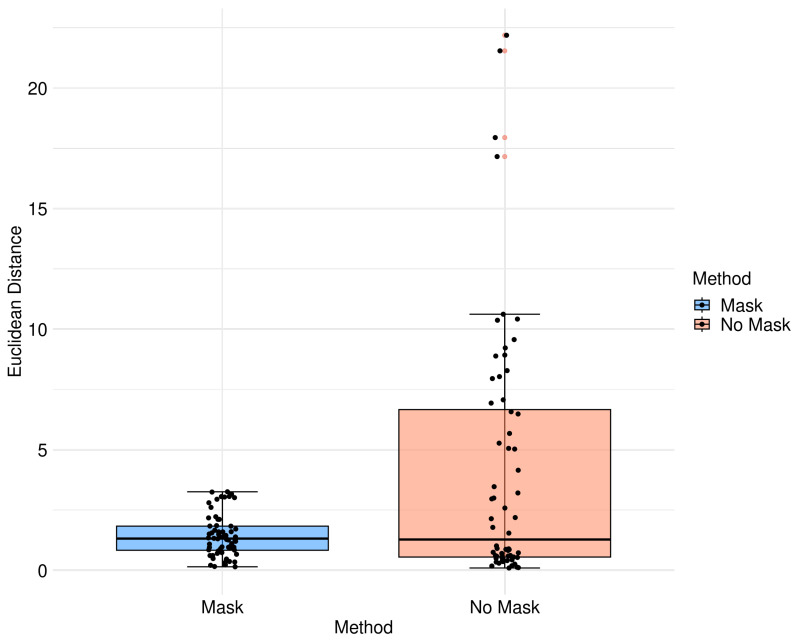
Box diagram of Euclidean distance for the different methods.

**Figure 11 diagnostics-13-03420-f011:**
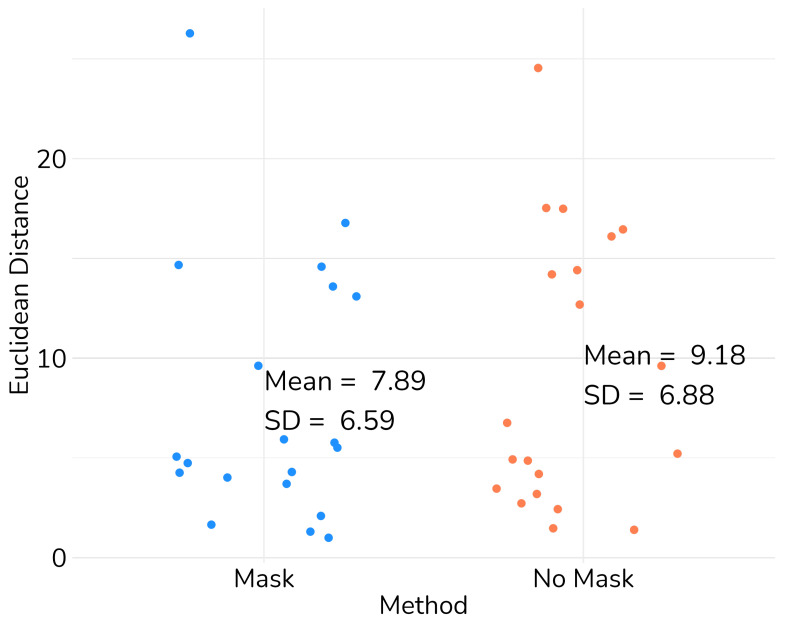
Scatter plot of the Euclidean distance between anatomical structures in the images from Imbanaco.

**Figure 12 diagnostics-13-03420-f012:**
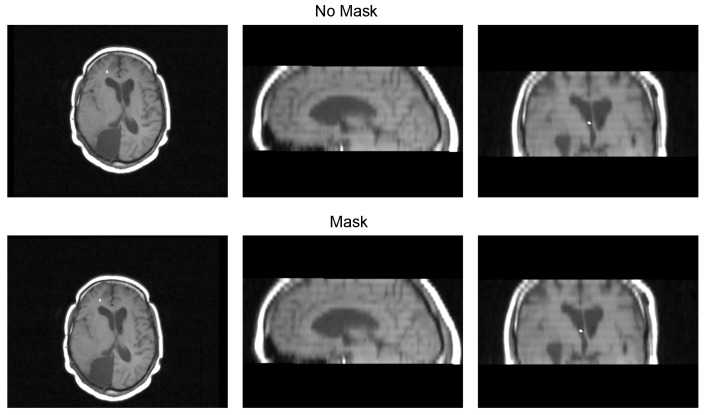
Image 3 fusion with no mask and with a mask.

**Figure 13 diagnostics-13-03420-f013:**
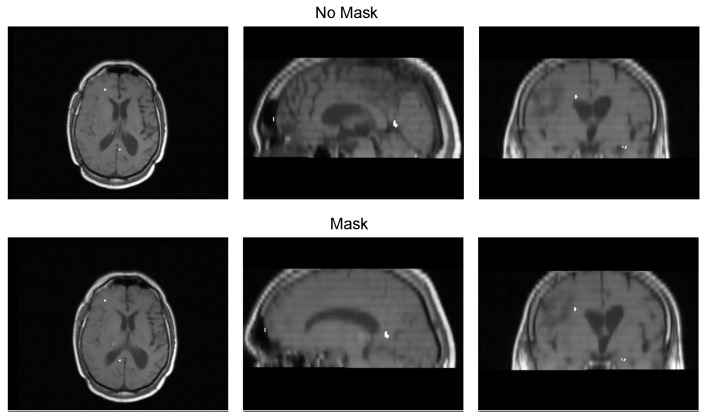
Image 6 fusion with no mask and with a mask.

**Figure 14 diagnostics-13-03420-f014:**
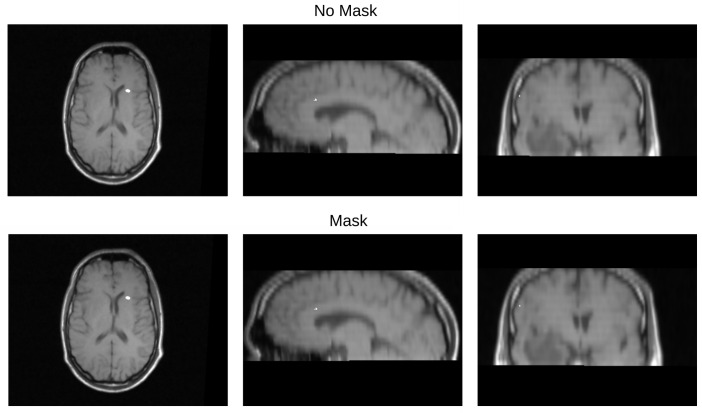
Image 8 fusion with no mask and with a mask.

**Table 1 diagnostics-13-03420-t001:** Original point positions in mm.

Point	X	Y	Z
1	0.0000	0.0000	0.0000
2	333.9870	0.0000	0.0000
3	0.0000	333.9870	0.0000
4	333.9870	333.9870	0.0000
5	0.0000	0.0000	112.0000
6	333.9870	0.0000	112.0000
7	0.0000	333.9870	112.0000
8	333.9870	333.9870	112.0000

**Table 2 diagnostics-13-03420-t002:** Reference resulting point for all eight images.

Image 1			Image 2			Image 3		
X	Y	Z	X	Y	Z	X	Y	Z
3.4167	−22.2013	−2.6957	2.9734	−29.8271	−17.7596	7.3801	−30.8327	−32.4198
331.6863	−22.0400	−3.7915	332.0465	−27.5075	−16.6379	333.7592	−29.2687	−27.2289
4.6098	309.1427	1.7573	2.2699	305.3511	−17.0817	9.9346	301.5044	−31.1614
332.8794	309.3039	0.6615	331.3430	307.6706	−15.9600	336.3137	303.0685	−25.9705
3.6098	−21.9385	107.9271	3.0988	−24.7405	94.2014	7.8149	−28.1302	73.5445
331.8794	−21.7772	106.8313	332.1718	−22.4210	95.3231	334.1940	−26.5661	78.7354
4.8029	309.4055	112.3801	2.3953	310.4376	94.8793	10.3693	304.2070	74.8029
333.0724	309.5667	111.2843	331.4683	312.7571	96.0010	336.7485	305.7711	79.9938
Image 4			Image 5			Image 6		
X	Y	Z	X	Y	Z	X	Y	Z
−4.4250	−22.1707	−6.5618	0.4343	−33.2795	−32.1174	−14.1576	−32.7423	−23.8992
327.7407	−21.7509	−8.7225	333.7756	−31.6846	−31.6177	308.7199	−34.2773	−26.0497
−4.2159	311.6391	−4.4026	1.3211	301.9878	−33.3221	−16.6352	298.2377	−21.9366
327.9498	312.0588	−6.5633	334.6624	303.5827	−32.8224	306.2424	296.7028	−24.0871
−4.0413	−22.2377	106.3093	0.9893	−30.9427	76.9555	−13.7173	−30.1729	85.4656
328.1243	−21.8180	104.1486	334.3306	−29.3477	77.4553	309.1603	−31.7078	83.3151
−3.8322	311.5720	108.4685	1.8761	304.3246	75.7508	−16.1948	300.8072	87.4282
328.3335	311.9917	106.3078	335.2174	305.9196	76.2505	306.6827	299.2722	85.2777
Image 7			Image 8					
X	Y	Z	X	Y	Z			
−7.6836	−35.2270	−19.1140	16.5968	−32.3464	−22.5351			
330.7646	−33.7368	−18.2985	337.2910	−39.1214	−20.8741			
−8.3642	304.4994	−16.5283	34.2465	299.5795	−22.6286			
330.0840	305.9896	−15.7128	354.9407	292.8044	−20.9676			
−7.7841	−37.0440	92.8582	16.5872	−26.0946	90.3929			
330.6641	−35.5538	93.6738	337.2813	−32.8696	92.0538			
−8.4647	302.6823	95.4439	34.2369	305.8313	90.2994			
329.9835	304.1725	96.2594	354.9310	299.0562	91.9603			

Coordinates in X, Y, and Z for the points obtained from applying the reference transform in 8 pairs of CT and MRI images from the RIRE dataset, without adding the synthesized electrodes. This point was used as a reference to compute the Euclidean distance in the registration procedure with the synthesized images with electrodes.

**Table 3 diagnostics-13-03420-t003:** Euclidean distance in mm for the image fusion methods.

Point	images 1		images 2		**images 3**		images 4	
	Mask	no Mask	Mask	no Mask	**Mask**	**no Mask**	Mask	no Mask
1	0.209	0.179	1.260	0.535	**2.114**	**3.469**	1.247	0.594
2	0.367	0.191	0.666	0.617	**1.553**	**4.152**	0.915	1.015
3	0.468	0.131	1.446	0.727	**1.307**	**6.576**	1.507	0.703
4	0.159	0.365	1.409	0.586	**1.279**	**7.070**	0.430	0.415
5	0.148	0.258	1.204	0.749	**1.714**	**5.029**	0.843	0.343
6	0.489	0.114	0.850	0.870	**0.987**	**5.274**	0.958	0.827
7	0.346	0.174	1.386	0.575	**0.960**	**8.033**	1.344	0.859
8	0.228	0.297	1.494	0.496	**0.972**	**8.282**	0.803	0.553
Point	images 5		**images 6**		images 7		**images 8**	
	Mask	no Mask	**Mask**	**no Mask**	Mask	no Mask	**Mask**	**no Mask**
1	3.064	2.998	**0.737**	**5.057**	1.327	0.909	**3.161**	**10.620**
2	3.059	2.964	**1.366**	**17.157**	0.954	0.892	**1.108**	**5.676**
3	2.612	2.584	**1.595**	**9.568**	0.328	0.436	**3.041**	**9.223**
4	1.601	1.543	**2.803**	**21.541**	0.614	0.098	**1.396**	**7.952**
5	2.175	2.189	**1.594**	**6.484**	1.089	0.598	**3.258**	**10.370**
6	1.832	1.782	**2.116**	**17.945**	0.615	0.546	**1.854**	**6.935**
7	3.247	3.209	**1.661**	**10.419**	0.705	0.611	**3.014**	**8.926**
8	2.223	2.141	**2.948**	**22.187**	0.899	0.404	**1.836**	**8.886**

Euclidean distance between the resulting points computed with the transformation from the proposed method using a sampling mask and the method with no sampling mask; images 3, 6, and 8 show a lower Euclidean reduction when a mask was used.

**Table 4 diagnostics-13-03420-t004:** Reference structures in CT image.

Structure	Image 1			Image 2		
	**X**	**Y**	**Z**	**X**	**Y**	**Z**
Sylvian Aqueduct	1.619	132.161	128.294	1.719	130.778	−390.919
Anterior commissure	0.079	149.323	131.294	0.31	145.902	−365.217
Right lens	30.48	227.407	121.907	30.951	230.123	−365.558
Left lens	−39.392	221.421	120.383	−38.422	226.1	−367.741
Pineal gland	1.399	132.381	139.294	2.155	125.729	−375.287
	**Image 3**			**Image 4**		
	**X**	**Y**	**Z**	**X**	**Y**	**Z**
Sylvian Aqueduct	−0.001	133.384	−411.389	6.398	149.516	−571.414
Anterior commissure	−0.425	170.072	−410.326	−0.19	170.766	−565.312
Right lens	29.609	225.447	−457.272	−0.743	234.271	−598.263
Left lens	−36.389	224.62	−453.821	−54.969	215.592	−584.7
Pineal gland	1.039	141.247	−398.385	9.875	147.018	−558.196

Coordinates in X, Y, and Z of the located structures in the four CT images.

**Table 5 diagnostics-13-03420-t005:** Structures in MRI registered with mask.

Structure	Image 1			Image 2		
	**X**	**Y**	**Z**	**X**	**Y**	**Z**
Sylvian Aqueduct	−0.497	135.501	131.223	−0.941	137.412	−374.917
Anterior commissure	−2.628	159.994	140.26	−2.127	163.213	−365.627
Right lens	31.231	226.372	126.957	34.232	237.695	−379.789
Left lens	−38.222	220.542	124.31	−36.479	233.46	−377.886
Pineal gland	−0.381	134.024	139.112	−0.327	131.274	−372.335
	**Image 3**			**Image 4**		
	**X**	**Y**	**Z**	**X**	**Y**	**Z**
Sylvian Aqueduct	−0.139	146.147	−404.701	3.085	150.405	−571.851
Anterior commissure	−2.833	169.993	−409.066	−0.279	173.761	−569.139
Right lens	28.638	225.725	−456.317	13.94	230.782	−603.875
Left lens	−37.283	224.27	−452.711	−46.124	228.076	−603.892
Pineal gland	0.002	143.884	−399.845	5.151	144.134	−566.054

Coordinates in X, Y, and Z of the located structures in the resulted fused images using a sampling mask in the registration procedure.

**Table 6 diagnostics-13-03420-t006:** Structures in MRI registered without mask.

Structure	Image 1			Image 2		
	**X**	**Y**	**Z**	**X**	**Y**	**Z**
Sylvian Aqueduct	−0.469	136.396	131.874	−0.435	134.905	−378.153
Anterior commissure	−2.491	160.081	140.798	−2.048	158.782	−365.235
Right lens	31.606	226.472	127.22	33.893	231.655	−370.27
Left lens	−38.372	220.462	124.4	−36.656	226.912	−371.566
Pineal gland	0.05	131.032	134.609	0.013	129.051	−375.979
	**Image 3**			**Image 4**		
	**X**	**Y**	**Z**	**X**	**Y**	**Z**
Sylvian Aqueduct	1.229	146.558	−405.057	3.03	151.037	−571.368
Anterior commissure	−0.905	169.695	−409.541	−0.098	173.599	−569.108
Right lens	29.558	226.22	−455.332	14.847	230.98	−603.507
Left lens	−36.204	224.27	−452.584	−45.157	228.144	−605.603
Pineal gland	1.401	141.552	−399.967	4.646	144.436	−565.84

Coordinates in X, Y, and Z of the located structures in the resulted fused images without using a sampling mask in the registration procedure.

**Table 7 diagnostics-13-03420-t007:** Euclidean distance between resulting registered images.

Structure	Image 1		Image 2		Image 3		Image 4	
	**Mask**	**no Mask**	**Mask**	**no Mask**	**Mask**	**no Mask**	**Mask**	**no Mask**
Sylvian aqueduct	5.925	**4.921**	**13.588**	17.526	**14.668**	14.41	3.696	**3.458**
Anterior commissure	14.583	**14.198**	**13.094**	17.487	**0.994**	2.719	**4.738**	4.86
Right lens	5.511	**5.209**	**5.762**	16.451	**2.089**	**1.39**	16.774	**16.101**
Left lens	4.254	**4.191**	**4.291**	12.683	**1.299**	1.468	26.282	**24.544**
Pineal gland	5.059	**2.429**	**4.013**	6.754	**1.651**	3.188	9.615	**9.612**

Euclidean distance between the reference structures with the structures from the resulting fused image in the methods using the mask and with no mask.

**Table 8 diagnostics-13-03420-t008:** Global fusion evaluation metrics in resulting registered images.

Metric	Image 1		Image 2		Image 3		Image 4	
	**Mask**	**no Mask**	**Mask**	**no Mask**	**Mask**	**no Mask**	**Mask**	**no Mask**
RMSE	7334.771	7334.767	7399.4	7537.772	6194.527	6207.718	7655.544	7657.312
PSNR	95.501	95.501	94.34	94.154	97.759	97.738	94.361	94.359
SSIM	0.893	0.893	0.87	0.9	0.774	0.775	0.853	0.853
MI	0.574	0.574	0.433	0.416	0.433	0.438	0.348	0.348

Global performance metrics for the fused images using the mask and without using the mask.

## Data Availability

The data and code supporting the findings of this study are openly available. The code used in this research has been made publicly accessible on GitHub at the following repository: https://github.com/andresprzh/SEEGFusion. This repository includes detailed documentation and source code relevant to the study. No new datasets were created or analyzed in this study; thus, data sharing beyond the provided code is not applicable to this article.

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
