# Peer review of "MRI and CT Fusion in Stereotactic Electroencephalography (SEEG)"

_diagnostics, 2023, doi:10.3390/diagnostics13223420_

Round 1

Reviewer 1 Report

The authors describe a method for MRI and CT fusion for SEEG depth electrode registration. This certainly is an important issue in the precision of implantation procedure and electrode anatomical position control. However, this paper raises several important questions about these essential concerns in patients who need intracranial recordings for presurgical planning of their epilepsy.

Multimodal image fusion is the starting point for surgical planning of most stereotactic procedures, and therefore needs to be as accurate as possible. Commercially available image fusion softwares are usually accurate enough for those procedures. The registration errors, pointed by the authors, are usually acceptable and can be corrected by intraoperative image control.

The presented image fusion method is robust and based on several previously described methods, although its validity remains questioned by the authors themselves in the results chapter. Based on image datasets of only 8 patients, they observe that the only difference between the two techniques (with and without fusion mask) concerns the dispersion of localizing errors, and state that “our results remain unconclusive due to the limited dataset used in the validation procedure”, and “the validation using anatomical structures does not yield conclusive results “ (page 9).

In contrast, tables 2 to 6 display a large number of millimetric and sub-millimetric measures that are difficult to interpret. As for example, what is the meaning of the values corresponding to anatomical structures (Tables 4 and 5) ?  Applied to the SEEG procedures, one must consider that sub-millimetric errors are negligible, since depth EEG recordings are provided by a bipolar signal acquisition from two adjacent electrode contacts, with a surface recording of 14 mm² (with a diameter of 0.8 mm, and 2 mm contact length and 1.5 mm space); this particular point is not discussed by the authors.

The described method is based on the fusion of pre-implantation MRI and post-implantation CT scan, and consequently not used for electrode implantation strategy but only for post-implantation control of each electrode anatomical real position ; this is an important issue since such control allows for correlations between depth EEG interictal and ictal signals and anatomical structures conditions the further surgical strategy in these patients ; this point is not even mentioned by the authors.

I would recommend the authors to enrich their study by more validation cases, in order to demonstrate the utility, the role in clinical practice and the reproducibiliy of their fusion method for either SEEG electrode implantation procedure or anatomical position control for adequate clinical correlations and surgical strategy in patients with focal intractable epilepsy.

Some minor English errors and inadequate terms.

Author Response

We left a document explaining how we answer the comments.

Reviewer 2 Report

Firstly, I would like to congratulate the authors on their interesting work.
However, some points need to be addressed:
1) I advise the authors to work with a neurologist to revise the text since many medical concepts should be revised. Starting from the abstract, for example, this sentence is incorrect since not necessarily drug-resistant epilepsy implies neurological surgery is needed. Many patients are not good candidates for surgery.: “Epilepsy is a common neurological disorder that causes spontaneous recurrent seizures. It is treated using anti-epileptic drugs, but this treatment is ineffective for about 25 % of patients, requiring the resection of the epileptogenic tissue.”
2) The abstract should be written in a single paragraph.
3) The authors should work with English editing services to improve the readability of the manuscript.
4) Was this research protocol new or was it reproduced from another study?
5) All of the statistical analyses should be thoroughly described as well as the
results and parameters.
6) Was any objective comparative analysis done besides the graph regarding
“mask” and “no mask” methods?
7) Was the protocol tested and validated in a smaller sample using statistical
analyses?
8) The discussion should be improved and expanded and authors should include an in-depth analysis of the literature including studies with similar objectives anddesigns.
9) All the limitations of the study should be included.
10) Please provide information about the source of the data, yea, duration of thestudy, setting of the study

Extensive editing required.

Author Response

(The authors gave the same response as above.)

Reviewer 3 Report

  The manuscript presented from Hinesterize et al entitled “MRI and CT Fusion in Stereotactic Electroencephalography(SEEG)” is interesting and original, however the text should be improved, in detail the introduction and discussione are really poors. The authors should extend the number of reference.

Author Response

(The authors gave the same response as above.)

Round 2

Reviewer 3 Report

The authors improved the quality of the manuscript following reviewer’s comments 

Author Response

We submitted a revised version of our text
